# Compacted Anthropogenic Materials as Backfill for Buried Pipes

**DOI:** 10.3390/ma14040717

**Published:** 2021-02-03

**Authors:** Andrzej Głuchowski, Raimondas Šadzevičius, Rytis Skominas, Wojciech Sas

**Affiliations:** 1SGGW Water Centre, Warsaw University of Life Sciences—SGGW, 02787 Warsaw, Poland; wojciech_sas@sggw.edu.pl; 2Institute of Hydraulic Engineering, Vytautas Magnus University Agriculture Academy, 53361 Kaunas, Lithuania; raimondas.sadzevicius@vdu.lt (R.Š.); rytis.skominas@vdu.lt (R.S.)

**Keywords:** anthropogenic material, constrained modulus, geotechnics, oedometric test, buried pipe, construction and demolition wastes, industrial solid wastes

## Abstract

Buried pipe design requires knowledge about the fill to design the backfill structure. The interaction between the backfill envelope and the pipe impacts the structural performance of the buried pipe. The backfill material and compaction level respond to the backfill’s overall strength and, therefore, for pipe-soil interaction. The strength of backfill material is described in terms of modulus of soil reaction *E’* and constrained modulus *E_ode_*. As the *E’* is an empirical parameter, the *E_ode_* can be measured in the laboratory by performing the oedometer tests. In this study, we have performed extensive oedometric tests on five types of anthropogenic materials (AM). Three of them are construction and demolition materials (C–D materials) namely, recycled concrete aggregate (RCA), crushed brick (CB), and recycled asphalt pavement (RAP). Two of them are industrial solid wastes (ISW) namely, fly ash and bottom slag mix (FA + BS) and blast furnace slag (BFS). The results of the tests revealed that AM behaves differently from natural aggregates (NA). In general, the *E_ode_* value for AM is lower than for NA with the same gradation. Despite that, some of AM may be used as NA substitute directly (RCA or BFS), some with special treatment like CB and some with extra compaction efforts like RAP or FA + BS.

## 1. Introduction

The failure and settlement of pipes structures is a subject of many geotechnical efforts involving designing and testing activities. This process needs prior studies to reveal the potential geotechnical parameters value of embedment and foundation soils for designed underground structures. The substitute for natural soils is artificial soils. The term artificial soils or man-made soils describes a wide range of anthropogenic materials (AM) whose properties cannot be directly driven from the natural soils in the same stress and physical conditions.

The physical, chemical, and mechanical properties need to be determined for each kind of AM. Those properties may vary significantly from the natural aggregates (NA). A great part of AM constitutes construction and demolition (C–D) materials. The C–D materials sources are reclaimed construction parts crushed after substituting them from the demolition debris, constituted from metal parts concrete, wood, and brick parts. These materials sometimes are separated before the crushing phase to produce more homogeneous aggregates as recycled concrete aggregate (RCA), crushed clay brick (CB), and recycled asphalt pavement (RAP). The optimal moisture content (OMC) and water absorption tests for two RCA sources from house and pavement redevelopment projects have shown that the differences in the C–D materials’ properties are dependent on the source of the material. The OMC for RCA-1 and RCA-2 was 9.0% (dry density *ρ_d_* = 2.21 g/cm^3^) and 12.8% (dry density *ρ_d_* = 1.81 g/cm^3^) respectively. The water absorption is 1.43% and 1.77%, respectively, indicating an increase in the water content with OMC [1]. The size of differences between the value of such properties as OMC and water adsorption is even larger for various C–D materials [2]. The most common material strength test is the California bearing ratio (CBR) test, which is an empirical bearing capacity test. The CBR test provides information about the quality of the pavement subbase and subgrade by comparison to the referring material for which the CBR is equal to 100%. The CBR bearing capacity requirements usually shows that the optimum CBR value for subbase is higher than 80%, and for subgrade higher than 40% [3]. The AM usually has the highest CBR value in unsoaked conditions, which is also true for the NA.

Nevertheless, these differences are negligible for some C–D materials. The CBR values may differ significantly for the same AM with a different origin [4]. The same observation as for C–D materials can be made for the second great source of AM, which is industrial solid wastes (ISW). ISW utilization ratio remains at a deficient level, and some efforts to stabilize a mix of ISW was conducted [5]. 

Among ISW, the inorganic part of the group is a source of potential pavement engineering materials. The wastes are coming from the engineering industry, which is blast furnace slag (BFS) and fly ash and boiler slag mix (FA + BS) [6].

The available literature concerning C–D materials and ISW that investigates geotechnical properties is still limited to the physical properties or basic strength characteristics for pavement engineering applications. These works indicate a promising application of such materials.

The oedometric tests are one of the geotechnical tests which give information about mechanical characteristics. One of the parameters that can be driven from the oedometric test is constrained modulus or oedometric modulus (*E_oed_*) [7,8], which is the ratio of stress for material under vertical axial load in restrained lateral conditions. The *E_oed_* parameter is used to predict the deflection and the bulking potential of flexible buried pipes [9,10]. These events are directly dependent on the soil stiffness quantified by constrained modulus. For the coarse-grained soils, the oedometric tests are conducted in larger molds than in Casagrande-type oedometers, suitable for fine-graded soils. Rowe and Barden developed the apparatus for coarse-grained soils with a diameter from 75 to 254 mm (BSI 1990) [11,12]. For such tests also CBR and Proctor molds are often used [13,14]. The constrained modulus is frequently used for soil-structure interaction analysis, mostly due to the simplicity of the oedometric test and a wide range of known values of *E_oed_* for natural soils [15].

In this article, we conducted a series of physical and mechanical tests to characterize the AM constrained modulus. The tests were conducted on the C–D materials: RCA, CB, and RAP and on the ISW materials, which are BSF and FA + BS. Additionally, we run tests on the NA to compare the test results.

## 2. Material and Methods

### 2.1. Constrained Modulus Measurement

Soil stiffness is the crucial element of pipe vertical deflection determination to applied loads. During the pipeline design, soil stiffness modulus determination is the most complicated part of the project. The soil modulus value depends on soil type, soil density, moisture content, lateral pressure distribution, and time [16]. The soil one-dimensional constrained modulus is the slope of the stress-strain curve and is calculated by (1):(1)$$E_{ode}=\frac{\sigma^{\prime }v}{\varepsilon_{v}},$$
where the **σ*’*_v_** is the effective vertical stress, and *ε_v_* is a vertical strain. The constrained modulus *E_ode_* can be used interchangeably with a modulus of soil reaction *E’*, which is directly used to design buried flexible pipe [17]. 

The pipe system design relies on a soil envelope with high stiffness to prevent the pipe deformations, especially in side fill and pipe haunch zone, to provide adequate pipe buckling strength. The well-designed soil envelope assures pipe stability under long term moisture changes and reduces the earth and life load carried by pipe wall [18]. 

For constrained modulus calculation, the oedometric tests are required. In this study, the modified oedometer consolidation test was adopted for the determination of the compressibility of the tested AM subjected to vertical loads. The results were used for constrained modulus for five types of AM and one type of NA. In this work, we tested compacted soil specimens with the use of Proctor and vibratory-hammer compaction technique. The large-scale oedometric tests were conducted in constant stress steps until the primary consolidation ceased. The important aim of modified oedometer tests was to obtain compressibility characteristics of the tested materials in the moisture content in which the samples were compacted.

Our specimens’ large-scale oedometric tests took place in the Proctor cylinder (*d* = 150 mm, *h* = 120 mm), supporting no lateral movements of the soil to assure constrain conditions. The following sequence of loading step were applied: 12.5, 25, 50, 100, 200, 400, 800 kPa (where 800 kPa stress level was performed for the test results analysis).

The compaction characteristics of tested samples were analyzed. The secant modulus, which is a one-dimensional constrained modulus, was calculated for the fixed effective vertical stress level. The example of calculation procedure presents Figure 1. The results of *E_ode_* calculation was compiled with the relative compaction *R_C_*, where, *γ_d,max_* is the maximum dry density from the compaction test and, which is defined as (2):(2)$$R_{C}=\frac{\gamma_{d}}{\gamma_{d,\;max}},$$

### 2.2. Physical Properties Test Results

We prepared the standard soil gradation curve for six soil types with fractions 0–20 mm based on the mass share. This type of soil gradation curve meets the requirements of ASTM D2321 Class II poorly-graded gravely sand with little fines (SP) [19] and in AASHTO M145 notation A1, A3 [20]. The grain composition presents Table 1. The gradation curve fulfills the requirements of Polish [21], English [22], and American [23] codes concerning highway construction purposes. Because such a mix is commonly available in the market and fulfills buried pipes requirements we decided to adjust soil gradation in this study to such a composition. 

The laboratory physical properties evaluation also included compaction tests to characterize the OMC and maximal dry density *ρ_d,max_* (see Figure 2). The vibratory hammer compaction gets higher *ρ_d,max_* results than the standard Proctor compaction tests. For the Proctor compaction tests, the OMC was on a higher level. The test results present more consistent characteristics with explicit OMC points. 

For NA, the compaction characteristic has a standard course with apparent maximal dry density at optimum moisture content in saturation ratio (S_r_) between 0.9 and 0.95. The compaction with a vibratory hammer results in a higher compaction degree than compaction with the Proctor method. Such characteristics may also be observed for BFS. For RCA, the compaction characteristic is similar to the NA except for air-dry samples where maximal dry density was observed. The air-dry density is in cases of CB, and RAP is a favorable state for compaction. For FA + BS, the compaction curve has similar to the cohesive soil pattern with optimum moisture content in a saturation ratio above 0.95. The Proctor and Vibro compaction tests conclusion can show that the AM compaction is most beneficial in the air-dry state or near full saturation state. From a practical point of view, the air-dry conditions are hard to achieve on a construction site. The compaction is recommended in near full saturation conditions (0.95 < S_r_ < 1.0).

The soil compaction in moisture content between the dry and wet states results in a lower dry density of AM. Therefore, before compaction effort, one needs to focus on the saturation state to achieve the highest possible soil dry density. It is worth noting that the laboratory compaction technique in reference to the field conditions is not directly comparable. The Vibro compaction produces a different kind of compaction condition where the high horizontal stress in the soil is an effect of the strong horizontal pulses. These horizontal stresses produce different side constraint conditions from the Proctor tests. 

## 3. Results

### 3.1. Natural Aggregate

The oedometric tests were conducted for NA to compare test results with the literature’s data. Figure 3 presents the constrained modulus value for fixed stress level versus relative compaction state for Proctor and Vibro compaction. The NA has a higher constrained modulus value for Vibro compaction than for the Proctor compaction technique. The highest *E_ode_* at *R_C_* = 1 was equal to 72.0 MPa for Proctor and 84.6 MPa for Vibro compaction.

For each stress level, we approximated a power function to compare the *E_ode_* for a range of relative compaction from 1.0 to 0.90. For the approximated functions, the coefficient of determination R^2^ for each stress level is presented in Appendix B. 

For buried pipe engineering, the range of vertical stresses from the geostatic load *P_sp_* is usually between 7.0 kPa to 36.0 kPa. For Class II NA soil type and compaction level 0.90 < *R_C_* < 0.95, the constrained modulus shall be higher than 10.34 MPa to 17.93 MPa, respectively, to relative compaction. In Figure 3, the constrained modulus value levels are presented. The red-colored areas represent a compaction zone in which the *E_ode_* does not meet the abovementioned requirements. The yellow area represents a moderate compaction zone where the constrained modulus is in the range of requirements. Finally, the green area shows a compaction zone in which *E_ode_* value is higher than the requirements. The tested NA meets the ASTM D2321 criteria in the case of vibro compaction. The Proctor compaction in low stress (*σ*’*_v_* < 100 kPa) and compaction level (*R_C_* < 0.94) test results indicate that the ASTM requirements are not meet. The reason for that may be the nature of the Proctor compaction where impact hammer blows are relaxing the top few millimeters of soil sample. Therefore, this phenomena cannot be transferred to the site conditions.

### 3.2. Recycled Concrete Aggregate

One of the well-investigated AMs is RCA. Numerous studies report excellent performance in terms of pavement engineering application [2,24,25,26]. The feasibility of using RCA in pavement engineering was conducted with the use of various tests. The CBR may vary from 36% to 160% where most of the research reports present CBR on the same level or higher in comparison to NA in the same test conditions [27,28]. RCA is produced during the crushing process. The crushing process and the resultant particle size impacts the soil’s mechanical performance [29]. The particle size of RCA impacts the amount of adhered mortar. The grains with fraction 4–8 mm have 33–65% of adhered mortar, and in the case of the fraction 8–16 mm, 23–44%. For particles in fractions of 0–0.3 mm, the adhered mortar constituted 65% of aggregates mass [30,31,32]. The water adsorption, especially in outer layers attached to the mortar, is the reason for the lower mechanical properties of RCA compared to NA [33,34]. RCA’s crushing value in comparison to NA is higher, and for fraction 5–10 mm the crushing value is equal to 45% and for fraction 10–20 mm it is 33% [35]. The adhered mortar and crushing-induced internal cracks impact the durability of RCA [36]. The properties of RCA are highly dependent on age, concrete strength, atmospheric exposition, etc. Some studies report that these properties may be lower than NA [37], nevertheless, the RCA, in general, is recognized as a mechanically sufficient base course substitute for NA [38]. For example, RCA has a higher resilient modulus and accumulates less plastic strain than the NA [39]. One of the methods that improve RCA chemical properties is CO_2_ treatment of aggregates, termed carbonization [40]. The carbonization process reduces the pH leachate [41]. The RCA constrained modulus value presents Figure 4. 

The value of the *E_ode_* in comparison to the NA is lower in compacted samples in high effective stress levels. Nevertheless, the RCA behavior at low-stress levels and lower com-paction has higher constrained stiffness than the NA. This phenomenon is clearly con-nected with higher roughness of the RCA surface, which results in additional resistance to the compression at low compaction and stress conditions. At higher stress levels, the forces between grains are high enough to overcome the effect of surface roughness.

Additionally, RCA in high compaction and stress conditions has higher crushing susceptibility. The adhered mortar is more likely to detach from the concrete aggregates. The Proctor compaction test gives much lower constrained modulus values than the Vibro compaction. The modulus value at low relative compaction (0.85 to 0.9) is much higher, even higher than for RCA compacted with Proctor’s method at RC = 1.0.

The ASTM requirements for Proctor compacted RCA samples are not meet at low-stress levels. The reason for that is as well as in the case of the NA, relaxed topsoil layer. Nevertheless, we can observe that this is not a case for Vibro compaction.

In certain conditions, Proctor compacted, RCA has higher constrained stiffness properties in lower-stress levels than NA. It is worth to note that RCA tends to increase its strength properties during the time after building in. The cementitious agents which are released after crushing cause self-cementation, and the increase of constrained modulus value can be expected [41].

### 3.3. Crushed Clay Brick

The CB is C–D material, usually deposited in landfills rather than used as earth backfill material. The CB use in concrete get comparable results of tensile and flexural strength and significantly reduced value of elasticity modulus [42,43]. The CB maximum dry density is lower than NA and is usually below 2.0 g/cm^3^ [4,43]. The OMC is higher than for NA and RCA and is between 10.7% and 14.9%. The frost resistance is significantly lower for CB than for NA and is equal to 55.8% (for NA 97.5%). The bearing capacity tests have shown that CB has CBR from 36% to 138% [4,43,44]. CB performs satisfactorily in low moisture content. Some studies show that the CB performs better when blended with other recycled aggregates, for example, RCA [44,45].

Figure 5 presents the constrained modulus characteristics for CB. The *E_ode_* characteristics have a high value at low stress and compaction level, similar to RCA characteristics.

In the case of CB, this relationship is even higher than for RCA. Low compacted CB (*R_C_* = 0.9) at *σ’_v_* equal to 12.5–50 kPa has a higher constrained modulus value than the NA. The high roughness of the CB surface causes this phenomenon, at high-stress levels, the CB particles suffer from the high internal stresses which impact internal porous structure. The high strength effect has much less impact, and finally, we can observe a drop in CB stiffness. What is more, the CB structure at lower stress levels may behave as preconsolidated due to compaction. The effect of low compaction on constrained modulus is high enough to give a higher *E_ode_* value at *σ’_v_* between 50 and 100 kPa at *R_C_* between 0.90 and 0.92 than in full compaction (*R_C_* = 1.0). This behavior was observed in the case of Vibro compaction. 

The ASTM requirements for the Proctor compaction at low *R_C_* level is generally not fulfilled. Nevertheless, in some cases, the low-stress level *E_ode_* value is higher. Considering the fact of the Proctor compaction disturbances at low-stress levels, CB may fulfill ASTM requirements. Another factor in favor of this statement is the grain roughness at low stress and compaction levels, which gives extra stiffness (see Figure 5a, where the 50 and 25 kPa points are in or above the moderate zone). The Vibro compaction is a less favorable method, as at low-stress levels, CB’s *E_ode_* does not meet the ASTM criteria. Higher compaction and higher crushing susceptibility might be a response to this phenomena.

### 3.4. Recycled Asphalt Pavement

The reports concerning RAP properties are limited, and only a few studies were conducted for this material, which is a significant product of recycled C–D waste [46]. The RAP strength properties usually do not meet the requirements concerning the bearing capacity. Therefore, the RAP-RCA blends were introduced to meet strength recommendations [47,48]. A significant improvement of strength properties was observed for RAP stabilized with cement addition [49,50]. Unfortunately, the addition of cement to RAP increases pH value and causes corrosion, which harms the environment [51]. The CBR value for RAP is between 30% and 35%. The RAP-FA stabilized mix studies have shown that the soil mix has a significantly higher CBR value than the untreated RAP (untreated RAP 19%, soil mix up to 94% after 28 days for field mixed soils) [52]. The evaluation of FA stabilization with recycled asphalt shingles has shown that this mix can be used as structural fill, the compressibility characteristics are similar to compacted sandy soil [53]. The addition of FA to the recycled pavement material increases the unsoaked CBR value significantly, at least three times, and increases the unsoaked CBR with increasing fly ash content [54]. The OMC is equal to 8.1% for RAP with 40% of fraction 5.6 mm–10 mm (SW in USCS classification). The water absorption is equal to 8.6% for coarse fraction and 22.4% for a fine fraction [47].

Figure 6 presents constrained modulus characteristics for RAP. The tests were performed in constant temperature conditions. The Proctor compaction of RAP shows that constrained modulus value at a low relative compaction state is slightly higher than *R_C_* = 1.0, which is similar to the CB behavior. Nevertheless, the reason for that is the viscous nature of asphalt attached to the aggregates. The Vibro compaction test results indicate a behavior similar to NA what means the influence of compaction technique on RAP performance. The Proctor test is an impact technique that relay on passing the compaction energy by hammer blows. The Vibro compaction is a technique that passes the compaction energy through horizontal micro-movements, which do not deform. These horizontal forces prevent the RAP from the grain deformation at higher vertical stress levels. Therefore, the Proctor compaction gives higher constrained modulus values at low relative compaction levels. The RAP *E_ode_* value at *R_C_* = 1.0 is comparable between Proctor and Vibro compaction. 

From the effective vertical stress level perspective, the constrained modulus value is significantly lower than for NA in the same conditions. The constrained modulus for *σ’_v_* = 12.5 kPa is significantly higher for low compacted RAP in the Proctor method. 

The ASTM requirements for RAP are not fulfilled in both compaction cases. It might be favorable to stabilize RAP with RCA or FA to achieve higher *E_ode_* values. Additionally, the temperature conditions are an important factor as well. Finally, RAP tends to increase their mechanical properties with time. The grain viscous deformation may produce better contacts between grains what positively impacts *E_ode_* value. Such behavior, however, is associated with increased material settlements.

### 3.5. Fly Ash and Boiler Slag Mix

FA + BS are part of coal combustion products (CCP). The laboratory studies report that the FA + BS are suitable material for earth fill with the mechanical properties close to the commonly used soils. The friction angle of FA + BS is depending on density and is in a wide range between 28° and 55° [55,56]. The OMC is between 12% to 34%, with the maximum dry in range 1.16 to 1.84 g/cm^3^ [57]. The column leaching test results indicate that none of the heavy metals exceeds the required regulatory levels, and only the chromium (Cr) level exceeds the maximum contaminant level in the first few pore volumes of flow, and then its concentration decreases below required safety levels. The preventive method to reduce leachate metals is a pump-and-treat method after the compaction phase [58,59]. Lots of effort was put into the subject of soil stabilization. The FA + BS addition to cohesive soils results in liquid and plastic limit decrease, increased CBR value, and unconfined compressive strength [60]. Tests on the FA mix with soil and cement have shown that the CBR value was between 26% to 140% [61]. Studies concerning fly ash and bottom ash (BA) mix revealed that the optimal mix is 70% FA and 30% BA. The CBR value was in range 26% to 120% [62].

In Figure 7, the results of constrained modulus characteristics for FA + BS are presented. The *E_ode_* characteristic is similar to the NA. The lower compaction conditions result in a lower modulus value. With the increase of the *R_C_* value, the *E_ode_* increases as well. The *E_ode_* value is also close to the NA value (around 60.0 MPa for *R_C_* = 1.0 and *σ’_v_* = 800 kPa). The only difference between NA and FA + BS behavior is that the *E_ode_* value is higher in the NA case for *R_C_* = 1.0 (especially in Vibro compaction). On the other hand, the constrained modulus value for low compacted FA + BS (*R_C_* = 0.85 to 0.90) is higher than for NA, this phenomenon is again due to higher grain roughness in comparison to NA.

The ASTM requirements for FA + BS are generally not meet for Vibro compaction. For this kind of method, the only compaction up to *R_C_* = 1 is recommended. The Proctor compaction shows that the FA + BS might be compacted with use of this method. Despite three tests where *E_ode_* was low, the Proctor impact compaction can be recommended as a primary laboratory compaction method.

### 3.6. Blast Furnace Slag

BFS is ISW, which comes from the iron manufacturing process. The smelting process separates iron ore, and lighter lime, silica, and alumina compounds, which chemically bounded creates BFS [63]. The BFS properties greatly rely on the cooling conditions after the smelting process. As in FA + BS, BFS is used for soil stabilization with a particular focus on expansive soils [64]. The cohesive soil and BFS mix lower the liquid and plasticity limits [65]. The internal friction angle of BFS is 39°–43°, and specific gravity is between 2.29 up to 3.35 [63,66,67]. The CBR tests on BFS have shown that the bearing capacity is dependent on soil arrangement and inter-particle forces like matric suction and dilatancy). The BFS CBR bearing capacity is between 65% to 160% for standard and modified Proctor compaction technique [67].

Figure 8, constrained modulus characteristics for BFS are presented. The BFS *E_ode_* value changes in a similar manner to the CB. The constrained modulus for higher effective vertical stress levels follows the same pattern as for NA, and the pattern changes for BFS in low stress and low relative compaction levels. Finally, at *R_C_* = 1.0, the *E_ode_* characteristics are comparable with *E_ode_* for NA.

The Vibro compaction impact on soil behavior differs from the Proctor. The *E_ode_* value is usually higher when compared to NA. However, BFS maximum modulus value does not achieve the NA modulus value. The BFS *E_ode_* characteristic is reversed. Low compacted BFS has a higher modulus value than a fully compacted one. The Proctor compaction test gets standard results. *E_ode_* increases with *R_C_* and effective vertical stress, but the constrained modulus value is lower in **R_C_** equal to 1.0 and higher for *R_C_* from 0.85 to around 0.95.

The ASTM requirements in the case of BFS, fulfilled in the case of vibro compaction. The Proctor compaction seems to be less favorable but more tests should be conducted. BFS constrained modulus value is at the same level as RCA or NA.

## 4. Discussion

### 4.1. Compaction Method Effect on E_ode_ Value

The compaction technique impacts the *E_ode_* value in all six soil type cases. The AM compacted by the Vibro method has a lower constrained modulus value. In the case of RCA, the *E_ode_* have a slightly lower value. A comparison between the compaction methods in terms of *E_ode_* value shows that for RCA and BFS, the Vibro compaction is more favorable. The Proctor method is favorable for FA + BS. In the case of RAP and CB, the Vibro compaction gives better results, but the compaction method’s effect is not so pronounced. The CB and RAP *E_ode_* characteristics show that this AM has significantly different compaction characteristics from the NA. The RAP tends to behave viscously, and CB shows a higher *E_ode_* at a low relative compaction level. Proctor compacted AM has a higher *E_ode_* at low *R_C_* in comparison to NA. Because the haunch fill of buried pipe for Class II material needs to be moderately compacted (0.90 < *R_C_* < 0.95), and to provide the pipe with support against the soil and traffic loadings [68], we recommend the Vibro compaction with the use of vibrating compactors and vibratory plates. The Vibro compaction creates horizontal stress, which is favorable in the case of haunch fill.

### 4.2. Relative Compaction Effect on E_ode_ Value

The compaction degree quantified by the *R_C_* parameter shows that usually, AM follows the pattern where higher *R_C_* indicated higher *E_ode_* value. AM tends to behave higher constrained modulus value at low relative compaction degree. This anomaly in comparison to NA is mainly due to another grain surface and structure characteristics. The RCA and CB have high surface roughness. The CB and BS have a porous nature, and RAP is a dolomite gain coated by viscous asphalt. Only BFS and RCA behave closely to NA. In the case of AM, the crushing phenomena also have a significant impact. At higher stress levels, the constrained modulus is significantly lower at *R_C_* = 1. The *E_ode_* value for loose AM (*R_C_* ≈ 0.90) tends to be higher at low vertical effective stress levels.

Finally, in Appendix A (Table A1, Table A2, Table A3, Table A4 and Table A5), we present a characteristic modulus value at stress range from 12.5 kPa to 400 kPa as a reference for designing pipe fill design.

### 4.3. Effective Vertical Stress Effect on E_ode_ Value

AM application for buried pipe engineering, typically is in the range of vertical stresses from the geostatic load *P_sp_* is usually between 7.0 kPa to 36.0 kPa. In this study, we assumed that the constrained modulus shall be in range of 10.34 MPa to 17.93 MPa for 0.90 < *R_C_* < 0.95 conditions (Class II NA soil type). In Figure 9, constrained modulus value at stress level from 12.5 to 50 kPa for Vibro compaction is presented for different AM. The red-colored areas represent a compaction zone in which the *E_ode_* does not meet the requirements. The yellow area represents a moderate compaction zone and, the green area shows a compaction zone in which *E_ode_* value is higher than the requirements. The ASTM line is the reference point which indicates the limit above which the tested soils have higher *E_ode_* than is expected in the ASTM code. The NA, RCA, and BFS fulfill the requirements. BFS in comparison to RCA and NA have lower E_ode_ but almost in all cases above ASTM reference line. What is more, the moderate compaction condition is beneficiary for this material as well in the CB case.

The CB is stiffer at a low compaction state. With the increase of compaction effort, the *E_ode_* value decreases to a limit between the yellow and red zone.

Therefore, the recommendation is to use this type of AS, as haunch fills in an uncompacted or slightly compacted state. The RAP and FA + BS show low constrained modulus value at such vertical effective stress conditions. To use RAP and FA + BS as a fill, we recommend compacting these soils with a higher effort using heavier equipment or a more extended compaction effort.

Finally, in Appendix A (Table A1, Table A2, Table A3, Table A4 and Table A5), we present a characteristic modulus value at stress range from 12.5 kPa to 400 kPa as a reference for designing pipe fill design.

## 5. Conclusions

In this study, we tested five types of anthropogenic material (AM), three of them are construction and demolition materials (C–D materials) namely, recycled concrete aggregate (RCA), crushed brick (CB) and recycled asphalt pavement (RAP), and two of them are industrial solid wastes (ISW) namely, fly ash and bottom slag mix (FA + BS) and blast furnace slag (BFS). For AM, a series of oedometric tests were conducted to evaluate constrained modulus *E_ode_* value, which is often used for buried pipe design. The tests were conducted in different relative compaction *R_C_* conditions using Proctor and Vibro compaction techniques with standard compaction effort. The stress range in the oedometric test was from 12.5 kPa to 800 kPa. The test results and analysis lead to the following conclusions:Vibrocompaction is a more efficient compaction technique. The AM dry density is usually higher when compared with the Proctor standard compaction method except for RAP. The laboratory vibrocompaction does not necessarily produce field conditions in a pipe trench.The optimal moisture content (OMC) occurs at an air-dry state for RCA, CB, RAP, and BFS. The same behavior is observed for NA. In the case of FA + BS, OMC is at saturation ratio (*S_r_*) around 0.95. This phenomenon also occurs for the rest of the tested AM. There are two optimal moisture states, dry and wet. Because the dry state is hard to achieve at the construction site, we recommend compacting AM at wet conditions (*S_r_* = 0.95). Moderate moisture content is unfavorable (except for CB) as the compaction degree impacts *E_ode_* value.The AM generally has a lower *E_ode_* value in comparison to NA with the same gradation. Nevertheless, like RCA or BFS, some of them can be used as a NA substitute with the same compaction requirements as the difference is not that great. CB is an AM where we observed a constrained modulus value reverse. CB has a higher *E_ode_* at a low compaction state because of the textural and structural properties.RAP and FA + BS show little usefulness for buried pipe fill. The *E_ode_* value at standard geostatic conditions and compacted state (*R_C_* = 1.0) still barely reach modulus requirements for haunch fill. For this type of AM, we recommend using heavier compaction equipment or conducting longer the compaction effort.The oedometric test results indicate that the AM to use for buried pipe design needs a preliminary laboratory test. Therefore, we recommend testing AM before their application on the field.

## Figures and Tables

**Figure 1 materials-14-00717-f001:**
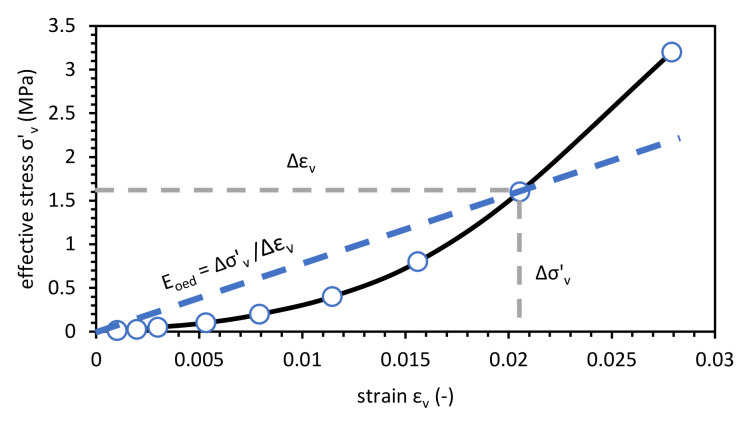
Secant modulus evaluation technique from oedometric tests on AM in fived effective vertical stress levels.

**Figure 2 materials-14-00717-f002:**
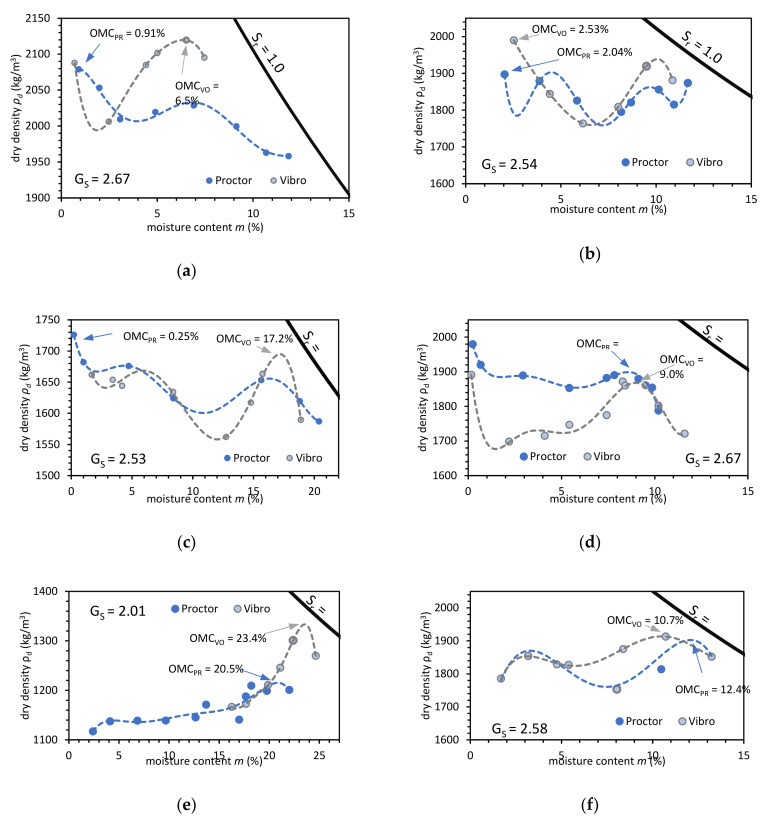
Compaction characteristics as a function of soil dry density versus moisture content of natural aggregate: (**a**); construction and demolition materials: (**b**); RCA; (**c**) CB; (**d**) RAP, and industrial solid wastes; (**e**) FA + BS; (**f**) BFS. S_r_ = 1.0 indicates full saturation conditions, G_S_ indicates specific gravity.

**Figure 3 materials-14-00717-f003:**
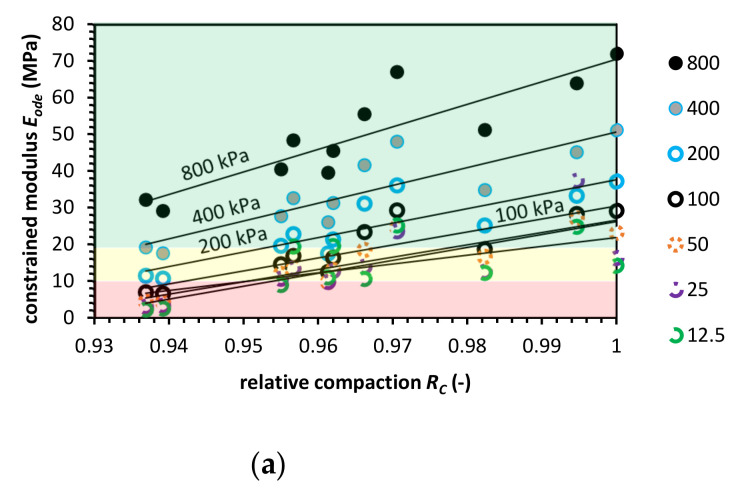
Constrained modulus value versus relative compaction value for natural aggregate (NA) for (**a**) the Proctor compaction and (**b**) the Vibro compaction method. Colored areas mean required constrained modulus value, colors indication: green-zone of fulfilled *E_ode_* modulus requirement; yellow-moderate zone, where the constrained modulus is in the range of requirements; red-zone in which the *E_ode_* do not meet pipe fill requirements.

**Figure 4 materials-14-00717-f004:**
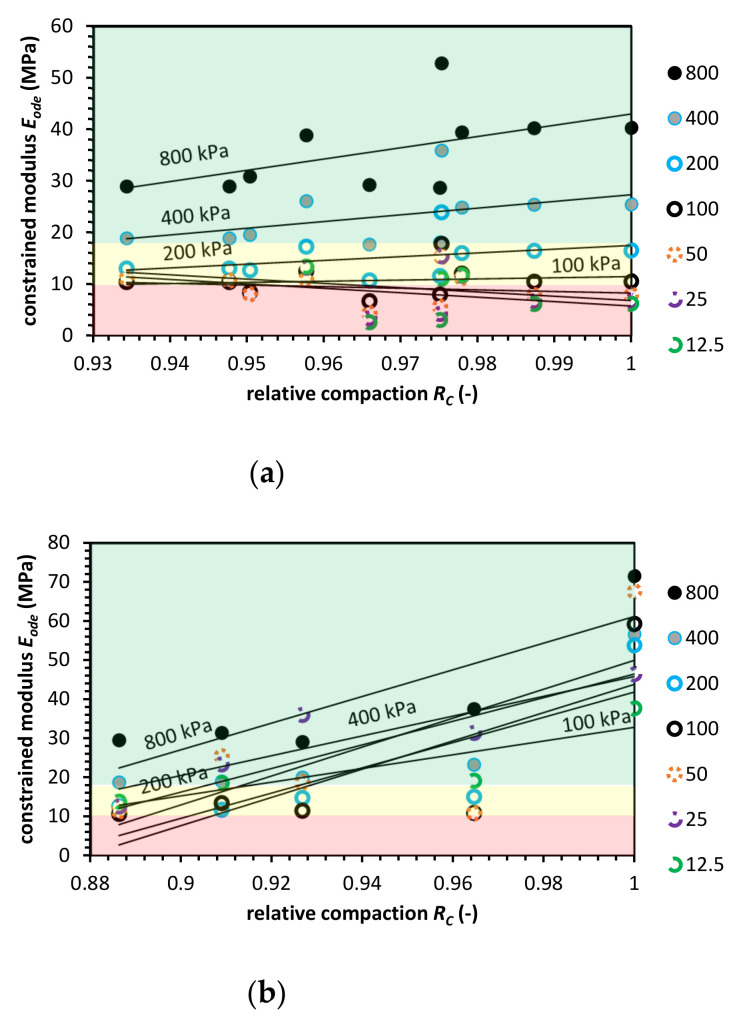
Constrained modulus value versus relative compaction value for recycled concrete aggregate (RCA) for (**a**) the Proctor compaction and (**b**) the Vibro compaction method. Colored areas mean required constrained modulus value, colors indication: green-zone of fulfilled *E_ode_* modulus requirement; yellow-moderate zone, where the constrained modulus is in the range of requirements; red-zone in which the *E_ode_* do not meet pipe fill requirements.

**Figure 5 materials-14-00717-f005:**
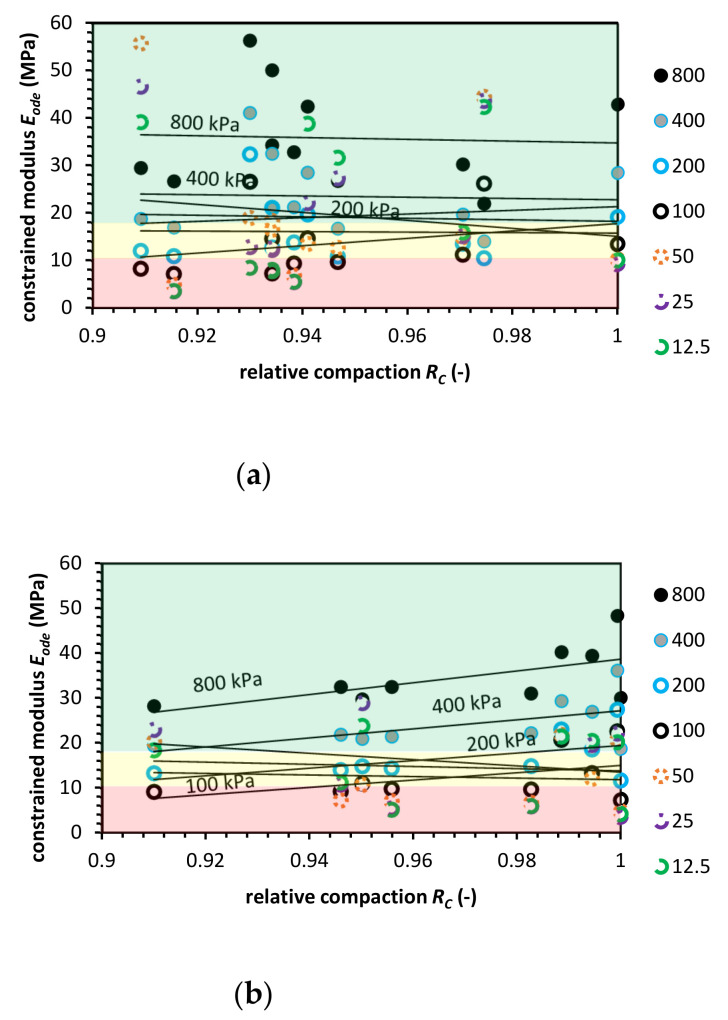
Constrained modulus value versus relative compaction value for recycled concrete aggregate (CB) for (**a**) the Proctor compaction and (**b**) the Vibro compaction method. Colored areas mean required constrained modulus value, colors indication: green-zone of fulfilled *E_ode_* modulus requirement; yellow-moderate zone, where the constrained modulus is in the range of requirements; red-zone in which the *E_ode_* do not meet pipe fill requirements.

**Figure 6 materials-14-00717-f006:**
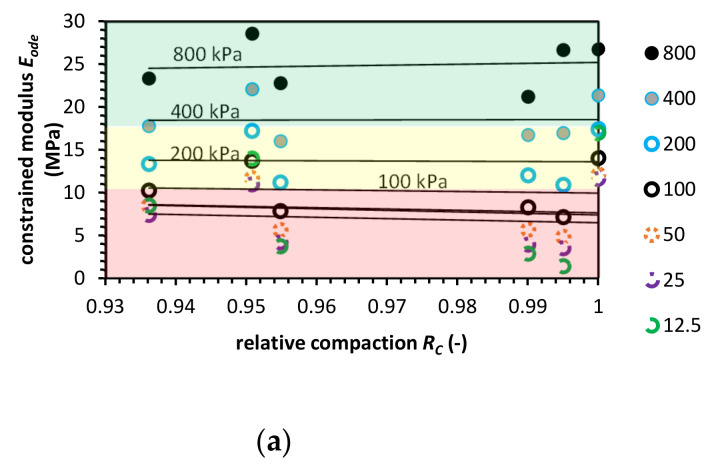
Constrained modulus value versus relative compaction value for recycled asphalt pavement (RAP) for (**a**) the Proctor compaction and (**b**) the Vibro compaction method. Colored areas mean required constrained modulus value, colors indication: green-zone of fulfilled *E_ode_* modulus requirement; yellow-moderate zone, where the constrained modulus is in the range of requirements; red-zone in which the *E_ode_* do not meet pipe fill requirements.

**Figure 7 materials-14-00717-f007:**
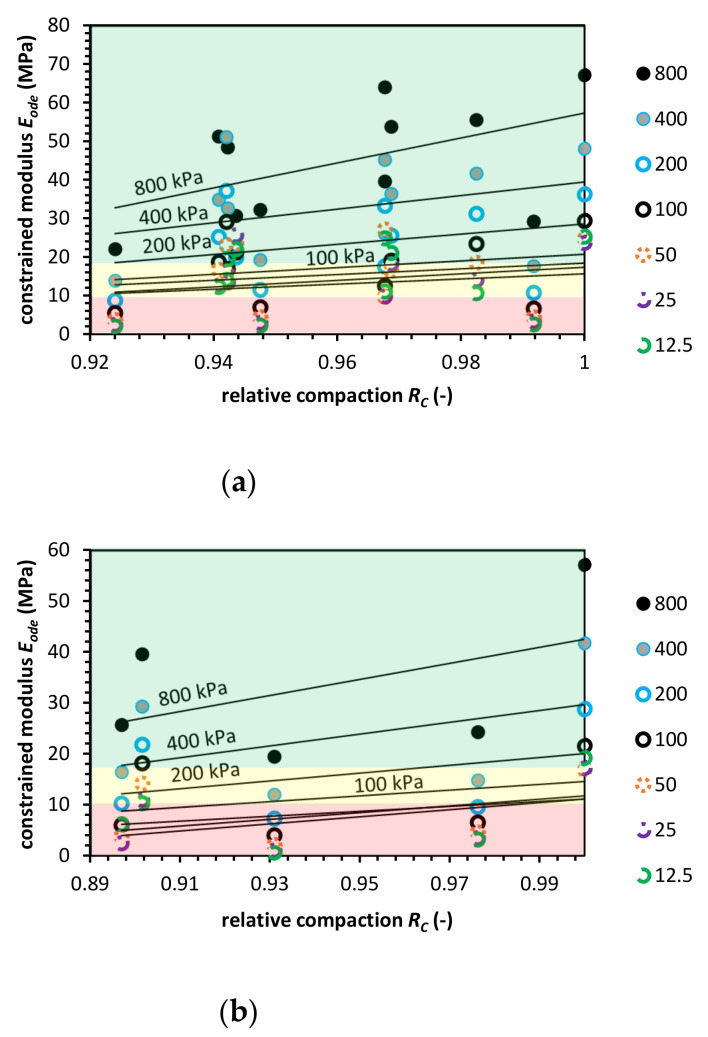
Constrained modulus value versus relative compaction value for fly ash and bottom slag (FA + BS) for: (**a**) the Proctor compaction and (**b**) the Vibro compaction method. Colored areas mean required constrained modulus value, colors indication: green-zone of fulfilled *E_ode_* modulus requirement; yellow-moderate zone, where the constrained modulus is in the range of requirements; red-zone in which the *E_ode_* do not meet pipe fill requirements.

**Figure 8 materials-14-00717-f008:**
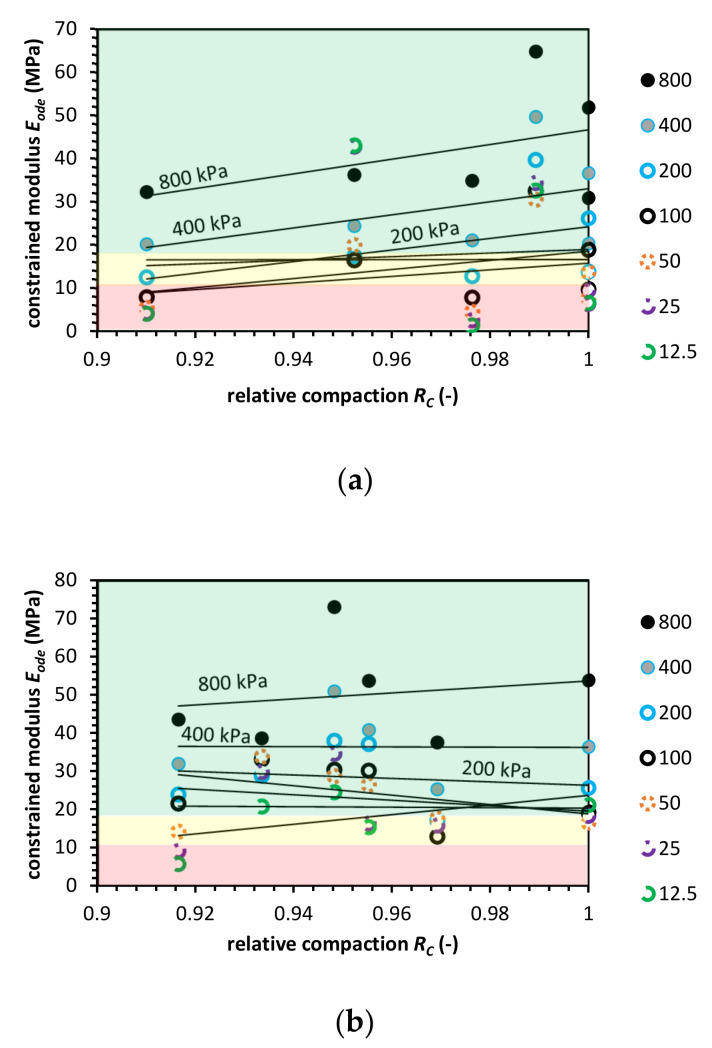
Constrained modulus value versus relative compaction value for blast furnace slag (BFS) for (**a**) the Proctor compaction and (**b**) the Vibro compaction method. Colored areas mean required constrained modulus value, colors indication: green-zone of fulfilled *E_ode_* modulus requirement; yellow-moderate zone, where the constrained modulus is in the range of requirements; red-zone in which the *E_ode_* do not meet pipe fill requirements.

**Figure 9 materials-14-00717-f009:**
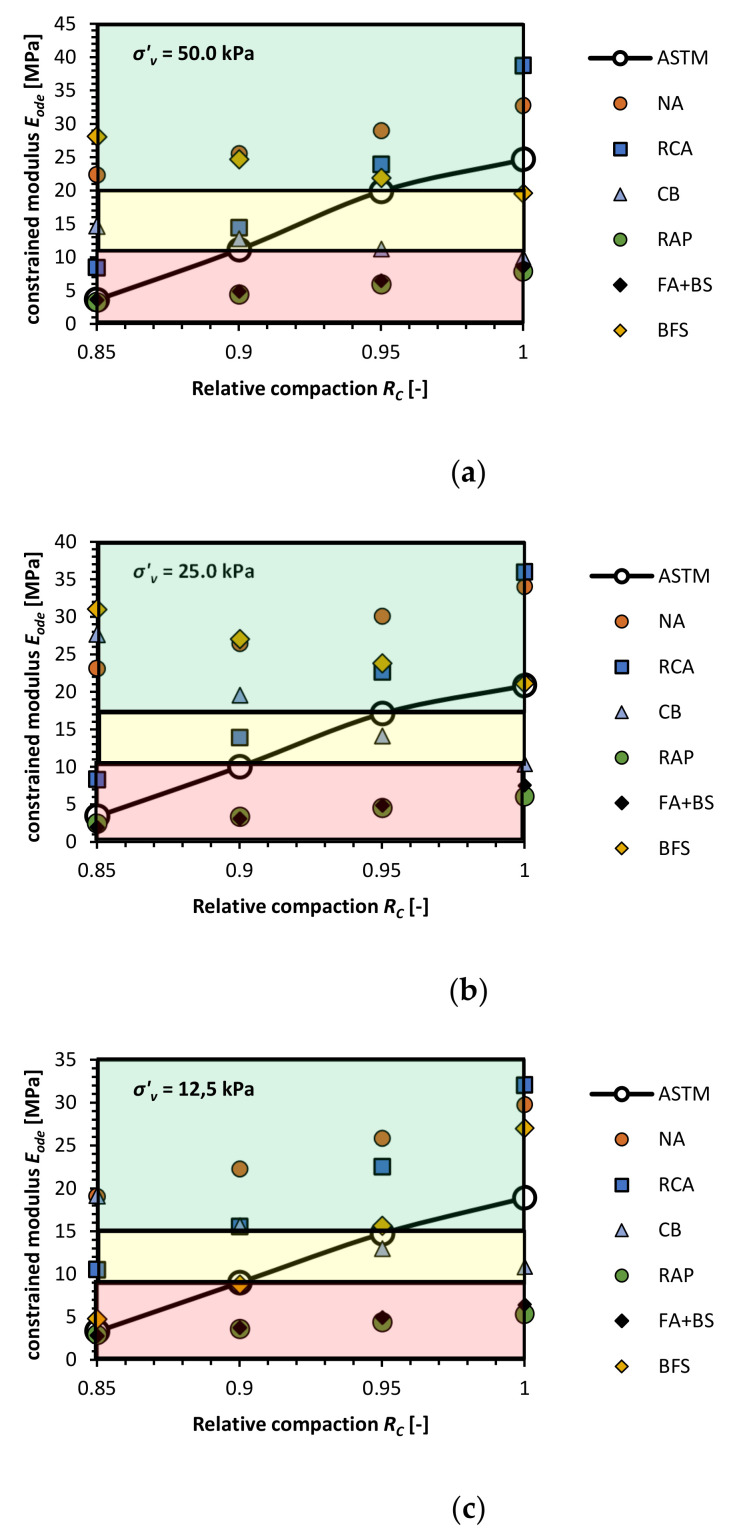
Constrained modulus value versus effective vertical stress at distinct stress levels for NA and five types of AS with denoted areas of required constrained modulus value, colors indicate: green-zone of fulfilled *E_ode_* modulus requirement; yellow-moderate zone, where the constrained modulus is in the range of requirements; red-zone in which the *E_ode_* do not meet pipe fill requirements, (**a**) *σ*’*_v_* = 50.0 kPa, (**b**) *σ*’*_v_* = 25.0 kPa, (**c**) *σ*’*_v_* = 12.5 kPa,.

**Table 1 materials-14-00717-t001:** Soil Gradation Curve Properties.

Sample Description	Value Range	Mean	Variance
D_10_ (mm)	0.1–0.19	0.224	0.045
D_30_ (mm)	0.4–0.61	0.462	0.00717
D_50_ (mm)	1.75–2.1	1.89	0.018
D_60_ (mm)	3.0–3.9	3.36	0.118
D_90_ (mm)	13.0–14.0	13.46	0.158
C_U_ (-)	20.5–30.0	-	-
C_C_ (-)	0.50–0.53	-	-
Sand content (%)	50.5–52.6	-	-
Fine content (%)	0.0–2.0	-	-

## Data Availability

Data is contained within the article.

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
