# Peer review of "Compacted Anthropogenic Materials as Backfill for Buried Pipes"

_materials, 2021, doi:10.3390/ma14040717_

Round 1

Reviewer 1 Report

This paper is significant because it is looking at alternative materials as buried pipe backfill as opposed to road base course or general fill.  The test methods are appropriate.  Certain aspects, however, questionable.  For instance, the applicability of vibrocompaction in a mold to the field conditions.  The side constraint is not the same.  Also use of high vertical stresses, e.g., 800 kPa is not necessarily relevant to buried pipe applications. There is an extensive literature review, however, significant work from the USA on RCA, RAP etc are not included resulting in biased conclusions about AM.  For instance, line 94 about lower mechanical properties for RCA relative to NA whereas there are many studies that showed the opposite.  Create more balanced literature survey.  A starting point could be Recycled Materials Resource Center papers and reports (rmrc.wisc.edu).  This study focuses on compaction and modulus; however long-term deformation and thermal  effects especially on RAP could be more decisive factors.

Specific Comments:

-define AS the first time it is used (line35)

-Section 2 is too long and as mentioned above does not reflect a balanced view of the literature

-RCA gains strength with time due to hydration of cement particles. There are many publications that show the opposite of Line 94.  References 25, 26 are not enough to make such a broad statement. 

-Refs 29 and 30 relate to carbonization and does improvement of chemical properties not physical properties.

-Give more justification of adjusting gradation to sand size (Line 170).  Is it a backfill requirement?

-Line 254:  Decrease of constrained modulus at 100 kPa from 50 kPa is probably an experimental variation not a basic behavior.  Furthermore call it "constrained modulus" and drop "soil" in the figures as these materials are not soil.

-Figures 3-14 are too busy and need not to be presented.  An example can be shown and Summary results can be presented like in Fig. 15.

-First conclusion about vibrocompaction (Line 405) should be removed or put in proper perspective.  The lateral constraint provided in a 155-mm mold is not necessarily to produce field conditions in a pipe trench.

Author Response

Respond for Review Report 1 for submission of a paper to Materials – MDPI journal

08.01.2021

Dear Reviewer,

We wish to thank You for all the remarks. Below are responses to Comments and Suggestions. 

“Certain aspects, however, questionable.  For instance, the applicability of vibrocompaction in a mold to the field conditions.  The side constraint is not the same.”

We agree with this remark. The compaction procedure is often a matter of researcher concerns when compared with in situ conditions. The laboratory impact Proctor compaction technique besides transmission of the compaction energy to the soil has little in common with the static roller compaction. We decided to strengthen these differences between laboratory and side compaction conditions in the methods chapter.

“Also use of high vertical stresses, e.g., 800 kPa is not necessarily relevant to buried pipe applications.”

Indeed, such high vertical stress for the buried pipe is too high. We presented these stress level results to perform modulus change analysis. Please note that in Appendix A we have shown only the values up to 400kPa. We appended an additional explanation in the Constrain Modulus Measurement chapter.

“There is an extensive literature review, however, significant work from the USA on RCA, RAP etc are not included resulting in biased conclusions about AM.  For instance, line 94 about lower mechanical properties for RCA relative to NA whereas there are many studies that showed the opposite.  Create more balanced literature survey.  A starting point could be Recycled Materials Resource Center papers and reports (rmrc.wisc.edu).  This study focuses on compaction and modulus; however long-term deformation and thermal  effects especially on RAP could be more decisive factors.”

Thank you for this remark. We balanced the literature survey, and we extended the description of RCA and RAP.

                “-define AS the first time it is used (line35)”

Done

                “-Section 2 is too long and as mentioned above does not reflect a balanced view of the literature”

We have rewritten Section 2, and now the section is part of section 3 – results.

“-RCA gains strength with time due to hydration of cement particles. There are many publications that show the opposite of Line 94.  References 25, 26 are not enough to make such a broad statement. 

We corrected our statement. We decided to show that, as Reviewer suggested, RCA, in general, has higher mechanical properties than NA, with a few exceptions.

“-Refs 29 and 30 relate to carbonization and does improvement of chemical properties not physical properties.”

We added information about carbonization positive impact on the RCA chemical properties

“-Give more justification of adjusting gradation to sand size (Line 170).  Is it a backfill requirement?”

The gradation curve fulfills the requirements of Polish, English, and American codes concerning highway construction purposes. Since such mix is commonly available in the market and meets buried pipes requirements, we decided to adjust soil gradation in this study to such composition. An explanation is given in chapter 4 in the manuscript.

“-Line 254:  Decrease of constrained modulus at 100 kPa from 50 kPa is probably an experimental variation not a basic behavior.  Furthermore call it "constrained modulus" and drop "soil" in the figures as these materials are not soil.”

We deleted the statement, and we changed the modulus name to the constrained modulus.

“-Figures 3-14 are too busy and need not to be presented.  An example can be shown and Summary results can be presented like in Fig. 15.

We deleted half of the figures. The rest, as an example for each AM were moved to chapter 3.

“-First conclusion about vibrocompaction (Line 405) should be removed or put in proper perspective.  The lateral constraint provided in a 155-mm mold is not necessarily to produce field conditions in a pipe trench.

We changed the conclusion, and we have included that the laboratory Vibrocompaction not necessarily produces field conditions.

We checked the manuscript for typos, and complete changes are presented in the manuscript. We show all changes in the red font style.

Thank you for your remarks on this manuscript.

Sincerely,

Andrzej, Raimondas, Rytis and Wojciech

Reviewer 2 Report

Dear Authors, please find my comments below:

1) Anthropogenic materials are firstly abbreviated in the manuscript as AM, then you use "AS" instead of AM in the rest of the study. Please unify the abbreviation throughout the paper, I advise using AM instead of AS. 

2) Delete the empty line 14 

3) Line 158 has an unnecessary line break in it

4) You probably wanted to write mass share in line 171, since you probably measured mass instead of weight. 

5) please cite the ASTM D2321 standard and AASHTO soil classification system in the reference as well.

6) Table 1 now contains the lower and upper limits for the measured 6 samples. I advise to show the average and standard deviation of the measured samples. Detailed results are welcomed also if they are open to the public in the Appendix.

7) Please add the a), b), c), etc. letters outside of the figures of Fig. 1, and the marking letter is sufficient, you wrote the rest in the caption of the figures. I also recommend to place to figures next to each other, and try to avoid that a single figure breaks into two pages. I have the same comment to all of the figures that contain multiple figures, such as Fig. 3, Fig. 4, etc.

8) Starting after line 218, values and units are often written together, and there is no space between the values and the unit of measure. 

8) the trend lines on constrained modulus often show poor fit in the figures, and linear lines do not show perfectly the trends of how the constrained modulus change due to Rc (if there is any), or if a measurement performed with huge deviations. Please discuss about the fitting of the trend lines and the deviations too in each figures' discussion. 

9) Starting from line 228, you wrote that the oedometric test results correspond with the ASTM, however, my eyes see pretty huge differences. Could you please expand these results in the paper a bit more detailed?

10) I also advise to put Fig. 4. on the same page or close to the discussion of it, now it placed in the middle of the discussion of Fig. 6. So please try to edit the discussion part that the Figures are close to the discussion of them, and not two pages away. 

11) There is no citation of Figure 9 and Figure 10 in the manuscript, where they were discussed.

12) Figure 15 does not have a), b) and c) markings. 

13) Overall, I find the discussion of the Constrained modulus and constrained soil modulus too few and too concise, do not compare the results in the figure enough (ASTM to AM, NA to AM), does not cover the variance of the results. Please improve the discussion part of the manuscript. Also, all the results of different AMs could be separated into individual subchapters. The editing of the presentation of the results a bit confusing now. I also think the discussion of a few findings could be earlier in the paper where you presented the results, but also it is welcomed since these subchapters are further discussion about the results in the following subchapters, which are also brief but gives valuable discussion comparing the results to each other due to different effects. 

Author Response

Respond for Review Report 2 for submission of a paper to Materials – MDPI journal

08.01.2021

Dear Reviewer,

We wish to thank You for all the remarks. Below are responses to Comments and Suggestions. 

“1) Anthropogenic materials are firstly abbreviated in the manuscript as AM, then you use "AS" instead of AM in the rest of the study. Please unify the abbreviation throughout the paper, I advise using AM instead of AS.”

We changed the abbreviation AS to AM.

“2) Delete the empty line 14 ”

Done

“3) Line 158 has an unnecessary line break in it.”

Done

“4) You probably wanted to write mass share in line 171, since you probably measured mass instead of weight”

Yes, we changed the word.

“5) please cite the ASTM D2321 standard and AASHTO soil classification system in the reference as well.”

Done

“ 6) Table 1 now contains the lower and upper limits for the measured 6 samples. I advise to show the average and standard deviation of the measured samples. Detailed results are welcomed also if they are open to the public in the Appendix.”

We added required information.

“7) Please add the a), b), c), etc. letters outside of the figures of Fig. 1, and the marking letter is sufficient, you wrote the rest in the caption of the figures. I also recommend to place to figures next to each other, and try to avoid that a single figure breaks into two pages. I have the same comment to all of the figures that contain multiple figures, such as Fig. 3, Fig. 4, etc.”

We changed the markings and captions as Rewiever suggested.

“8) Starting after line 218, values and units are often written together, and there is no space between the values and the unit of measure. ”

We have separated the units of measure.

“8) the trend lines on constrained modulus often show poor fit in the figures, and linear lines do not show perfectly the trends of how the constrained modulus change due to Rc (if there is any), or if a measurement performed with huge deviations. Please discuss about the fitting of the trend lines and the deviations too in each figures' discussion. ”

According to the second rewove we were advised to simplify the analysis so we removed the Figures from the manuscript.

“9) Starting from line 228, you wrote that the oedometric test results correspond with the ASTM, however, my eyes see pretty huge differences. Could you please expand these results in the paper a bit more detailed?”

We added the ASTM comparison chapter in the manuscript.

“-10) I also advise to put Fig. 4. on the same page or close to the discussion of it, now it placed in the middle of the discussion of Fig. 6. So please try to edit the discussion part that the Figures are close to the discussion of them, and not two pages away. “

We edited the manuscript so the figure is always after the description.

                “11) There is no citation of Figure 9 and Figure 10 in the manuscript, where they were discussed.”

Done

                “12) Figure 15 does not have a), b) and c) markings.”

We added the markings

“13) Overall, I find the discussion of the Constrained modulus and constrained soil modulus too few and too concise, do not compare the results in the figure enough (ASTM to AM, NA to AM), does not cover the variance of the results. Please improve the discussion part of the manuscript. Also, all the results of different AMs could be separated into individual subchapters. The editing of the presentation of the results a bit confusing now. I also think the discussion of a few findings could be earlier in the paper where you presented the results, but also it is welcomed since these subchapters are further discussion about the results in the following subchapters, which are also brief but gives valuable discussion comparing the results to each other due to different effects.”

We have modified the manuscript. We added the results and discussion chapter the article. We have extended the discussion chapter according to the reviewer suggestion.

We checked the manuscript for typos, and complete changes are presented in the manuscript. We show all changes in the red font style.

Thank you for your remarks on this manuscript.

Sincerely,

Andrzej, Raimondas, Rytis and Wojciech

Round 2

Reviewer 2 Report

Dear Authors, the revised paper is improved and presents the results better, however I has three comments that needs to be adressed before acceptance. 

1) Please check the meaning and correctness of the newly added texts, I can't really understood what author means under e.g. "The impact Proctor compaction, however, smiles the field compaction." or "which smiles the side vibro compaction conditions." or "Vibrocompaction smiles the field conditions"... I don't think any compaction needs to smile, please correct these sentences. 

2) You wrote in the reaction to a comment: 

     “12) Figure 15 does not have a), b) and c) markings.”

"We added the markings"

However, I don't see Fig. 15 at all in the manuscript. It is deleted from the revised version of the paper, as well as the discussion of the effective vertical stress effect on Eode value (chapter 4.3 in the first version of the paper). Why did you deleted this part of the paper? I see that it is beneficiary to delete a few figures showing poor fits and large variance of the results, and the paper was a bit too long and repetitive on those. But I don't get why this comparison was removed from the paper during revision. I advise to include it in the paper, since these small stress levels are not visilbe in Fig. 3-8. as well as this is a good direct comparison between the different AMs, since Fig. 3-8. does not compare the AMs to each other directly. 

3) I still advise to put R^2 to Fig. 3-8. to show the coefficient of determination of the linear regression trend lines. 

Author Response

Respond for Review Report 2 for submission of a paper to Materials – MDPI journal

26.01.2021

Dear Reviewer,

Once again, thank You for remarks, we applied them to the manuscript. Below we addressed specific explanations to Comments and Suggestions. 

“1) Please check the meaning and correctness of the newly added texts, I can't really understood what author means under e.g. "The impact Proctor compaction, however, smiles the field compaction." or "which smiles the side vibro compaction conditions." or "Vibrocompaction smiles the field conditions"... I don't think any compaction needs to smile, please correct these sentences. ”

We changed abovementioned issue concerning smiling. The new text is highlighted in manuscript with res style font.

“2) You wrote in the reaction to a comment: 

     “12) Figure 15 does not have a), b) and c) markings.”

"We added the markings"

However, I don't see Fig. 15 at all in the manuscript. It is deleted from the revised version of the paper, as well as the discussion of the effective vertical stress effect on Eode value (chapter 4.3 in the first version of the paper). Why did you deleted this part of the paper? I see that it is beneficiary to delete a few figures showing poor fits and large variance of the results, and the paper was a bit too long and repetitive on those. But I don't get why this comparison was removed from the paper during revision. I advise to include it in the paper, since these small stress levels are not visilbe in Fig. 3-8. as well as this is a good direct comparison between the different AMs, since Fig. 3-8. does not compare the AMs to each other directly.”

During the manuscript reediting we combined two reviews opinion. In some point, we decided to delete some figures but we did not updated this remark for what we apologize. Indeed, the discussion is interesting and we added it to the manuscript.

“3) I still advise to put R^2 to Fig. 3-8. to show the coefficient of determination of the linear regression trend lines. ”

We added the table with R2 in Appendix B

Thank you for your remarks on this manuscript.

Sincerely,

Andrzej, Raimondas, Rytis and Wojciech